# Influence of the Preoperative Duration of Symptoms on Patients’ Clinical Outcomes after Minimally Invasive Surgery-Transforaminal Lumbar Interbody Fusion for Degenerative Lumbar Spinal Diseases

**DOI:** 10.3390/medicina59010022

**Published:** 2022-12-22

**Authors:** Yoshiaki Hiranaka, Shingo Miyazaki, Takashi Yurube, Kohei Kuroshima, Masao Ryu, Shinichi Inoue, Kenichiro Kakutani, Ko Tadokoro

**Affiliations:** 1Department of Orthopaedic Surgery, Anshin Hospital, Kobe 650-0047, Japan; 2Department of Orthopaedic Surgery, Kobe University Graduate School of Medicine, Kobe 654-0142, Japan

**Keywords:** lumbar degenerative disease, minimally invasive treatment, duration of symptoms, lower back pain, Oswestry disability index, visual analogue scale

## Abstract

*Background and Objectives*: The impact of the duration of symptoms (DOS) on postoperative clinical outcomes of patients with degenerative lumbar spinal diseases is important for determining the optimal timing of surgical intervention; however, the timing remains controversial. This prospective case–control study aimed to investigate the influence of the preoperative DOS on surgical outcomes in minimally invasive surgery-transforaminal lumbar interbody fusion (MIS-TLIF). *Materials and Methods*: Patients who underwent single-level TLIF for lumbar degenerative diseases between 2017 and 2018 were reviewed. Only patients with full clinical data during the 1-year follow-up period were included. The patients were divided into two groups (DOS < 12 months, group S; DOS ≥ 12 months, group L). The clinical outcomes, including the Oswestry disability index (ODI) and visual analog scale (VAS) for lower back pain, leg pain, and numbness, were investigated preoperatively and at 1, 3, and 6 months, as well as 1 year, after surgery. Furthermore, postoperative patient satisfaction 1 year after surgery was also surveyed. *Results*: A total of 163 patients were assessed: 60 in group S and 103 in group L. No differences in baseline characteristics and clinical outcomes were found. The ODI and VAS significantly improved from the baseline to each follow-up period (all *p* < 0.01). Group S had significantly lower ODI scores at 3 months (*p* = 0.019) and 6 months (*p* = 0.022). In addition, group S had significantly lower VAS scores for leg pain at 3 months (*p* = 0.027). In a comparison between both groups, only the patients with cauda equina symptoms showed that ODI and leg pain VAS scores at 3 months after surgery were significantly lower in group S (19.9 ± 9.1 vs. 14.1 ± 12.5; *p* = 0.037, 7.4 ± 13.9 vs. 14.7 ± 23.1; *p* = 0.032, respectively). However, the clinical outcomes were not significantly different between both groups 1 year after surgery. Patient satisfaction was also not significantly different between both groups. *Conclusions*: Patients with a shorter DOS tended to have a significantly slower recovery; however, clinical outcomes 1 year after surgery were good, regardless of the DOS.

## 1. Introduction

Surgical treatment for lumbar degenerative diseases is expected to result in short- or long-term recovery from leg, back, or buttock pain and walking disability [1]. Various factors affecting surgical treatment outcomes have been reported to date. The preoperative duration of symptoms (DOS) is one of the factors affecting surgical treatment outcomes for lumbar degenerative diseases. The DOS is considered clinically important because conservative treatment of lumbar degenerative diseases is often successful [2]; however, prolonged symptomatic neurological compression can cause inferior surgical outcomes. Ng et al. [3] reported that a DOS >33 months had less favorable outcomes, and Johnsson et al. [4] reported more favorable surgical outcomes in patients with a DOS of <4 years. In contrast, Movassaghi et al. [5] reported that the DOS did not significantly affect postoperative clinical outcomes, reoperation rates, or patient satisfaction. Thus, the impact of DOS on clinical outcomes remains controversial, and the optimal timing of surgical intervention remains unclear. Moreover, most studies that have reported on the impact of DOS do not have fixed numbers of surgical levels or procedures.

In recent years, various minimally invasive surgeries (MISs) have been reported [6]. One of them, MIS transforaminal lumbar interbody fusion (MIS-TLIF), is an effective treatment for symptomatic lumbar degenerative diseases. MIS-TLIF reduces tissue damage, muscle retraction, and skin incisions using tubular retractors via a muscle-splitting approach as an alternative surgery to the conventional open approach. In addition, it reduces complications associated with open lumbar spinal fusion surgery [6]. MIS-TLIF has also been reported to reduce intraoperative blood loss, immediate postoperative pain, hospitalization stay, and infection rates compared to open TLIF [7,8,9]. Furthermore, Lv et al. showed that patients who underwent MIS-TLIF had reduced back pain and better Oswestry disability index (ODI) scores compared to open TLIF at 6, 12, and 36 months postoperatively [10]. However, even with the advent of this minimally invasive and stable performance surgery, there is no consensus on the optimal timing for surgical treatment. Thus, this prospective study aimed to ascertain the effect of preoperative DOS on the surgical outcomes of MIS-TLIF over a 1-year follow-up period.

## 2. Materials and Methods

### 2.1. Patients and Methods

The Institutional Review Board of Anshin Hospital approved the study protocol (IRB No. 117). We prospectively investigated patients who underwent MIS-TLIF for lumbar degenerative disease between March 2017 and June 2018. The target diseases were lumbar degenerative diseases, including lumbar spinal stenosis (LSS) with lumbar degenerative spondylolisthesis (DS), LSS without DS, and isthmic spondylolisthesis (IS). Only patients who were able to be treated conservatively from the beginning in our hospital after the onset of symptoms were included in this study. All patients underwent a standardized trial of nonoperative and conservative treatments, including medication, physical therapy, and pain block, for at least 3 months before surgery. Surgical intervention was considered after the failure of conservative treatment. The number of surgical levels was only a single level. Patients who had undergone a previous lumbar surgical procedure (decompression and/or fusion) were excluded. We also excluded patients who were followed up with for <1 year or had reoperation within 1 year after MIS-TLIF. Thus, this study included 188 consecutive patients.

Patients were prospectively followed up at 1, 3, and 6 months, as well as 1 year, postoperatively. The recorded preoperative parameters included sex, age, body mass index (BMI), diabetes mellitus (DM) history, smoking history (Brinkman index ≥400), preoperative paralysis (any of the lower extremity manual muscle testing ≤3), types of neurological symptoms (cauda equina syndrome plus mixed-type or radiculopathy), postoperative radiographic parameters (pelvic incidence minus lumbar lordosis and pelvic tilt), and DOS (months). DOS was defined as the period from the onset of neurological symptoms to surgery. In addition, based on the reports of Movassaghi et al. [5] and Radcliff et al. [11], patients were grouped into the short-term DOS group (group S), with a DOS <12 months, and the long-term DOS group (group L), with a DOS >12 months. Clinical outcomes were assessed according to the Oswestry disability index (ODI) and visual analog scale (VAS) for lower back pain, leg pain, and numbness preoperatively and at 1, 3, and 6 months, and 1 year postoperatively. The ODI and VAS changes were defined as preoperative ODI minus postoperative ODI and preoperative VAS minus postoperative VAS, respectively. We also evaluated the patient satisfaction score using the VAS (graded as ‘0’ for unsatisfied to ‘100’ points for very satisfied) at 1 year postoperatively. In addition, the patient satisfaction score was also evaluated based on two parameters: satisfaction with the surgery and satisfaction with the present condition (Figure 1).

### 2.2. Surgery

The criteria for MIS-TLIF included the following: (1) LSS combined with DS, (2) LSS combined with severe intervertebral disc degeneration, (3) LSS combined with foraminal stenosis, and (4) isthmic spondylolisthesis (IS).

Two senior surgeons performed the surgery. Patients were placed in a prone position under general anesthesia, and marking was performed preoperatively using an image intensifier. The surgeon stood on the more symptomatic side, and an approximate 3 cm incision was made at the level of the facet joint to be resected. The soft tissue was bluntly dissected using a 26 mm tubular retractor inserted to visualize the facet joint complex. The facet capsule was opened, and the inferior facet was resected using a high-speed burr. Then, the superior facet was resected from the tip to the superior border of the pedicle. Next, the ligamentum flavum was resected from the lateral to the medial side to expose the disc within the foramen. After removing the ligamentum flavum, the disc was exposed to the floor of the spinal canal. Then, the nucleus pulposus and cartilage from the vertebral endplates were completely removed to expose the bony endplates. The morselized bone graft was then packed into the anterior portion of the discectomy space. One of two structural implants of appropriate size, filled with additional bone graft, was impacted into the discectomy space. When necessary, the contralateral ligamentum flavum was resected to expose the exiting and traversing nerve roots. Then, instrumentations, including percutaneous pedicle screws (PPS) and rods, were inserted percutaneously, and the slip was corrected using the PPS system, as necessary. Finally, the set screws were used to tighten the rods.

During the postoperative therapy, all patients wore hard corsets for 2 months and underwent the same conservative protocol, which included activity modification, anti-inflammatory medications, and physical therapy.

### 2.3. Statistical Analysis

The collected data were statistically analyzed using the SPSS software (version 20.0; SPSS, IBM Corporation, Armonk, NY, USA). Data are shown as mean ± standard deviation. For statistical analysis, sequential changes in the ODI, lower back pain, leg pain, and numbness VAS scores from the preoperative baseline to 1 year after surgery in groups S and L were analyzed using a paired *t*-test. The baseline patient demographics were compared using an unpaired *t*-test. In addition, we used unpaired *t*-tests to compare the ODI and VAS at the baseline and follow-up period, ODI and VAS changes from the baseline to the follow-up period, and patient satisfaction scores between groups S and L. All statistical tests were two-sided. Statistical significance was defined as *p* < 0.05.

## 3. Results

Of the 188 patients who underwent MIS-TLIF for lumbar degenerative disease, 2 underwent reoperation within 1 year due to adjacent segmental diseases, and 23 could either not be followed up with for 1 year or had missing clinical outcome values at any time during the follow-up period. Finally, 163 patients were analyzed; group S comprised 60 patients, and group L included 103 patients. There were 79 male and 84 female patients, and the mean age at surgery was 67.4 years. Sixty-one patients had LSS without DS, 155 had LSS with DS, and 13 had IS. The follow-up rate at 1 month, 3 months, 6 months, and 12 months postoperatively were 99.4%, 97.3%, 91.5%, and 86.7%, respectively. Preoperatively, there were no significant differences in the baseline patient demographics between groups S and L (Table 1).

In both groups S and L, the ODI and lower back pain, leg pain, and numbness VAS scores showed significant improvement from the preoperative baseline to any time points after surgery (all *p* < 0.01) (Figure 2).

The ODI and lower back pain, leg pain, and numbness VAS scores at baseline and 1 year after surgery were not significantly different between groups S and L. However, ODI and leg pain VAS scores at 3 months postoperatively and ODI at 6 months postoperatively were significantly lower in group S (10.1 ± 8.9 vs. 14.0 ± 12.2; *p* = 0.019, 7.6 ± 14.1 vs. 13.9 ± 21.9; *p* = 0.027, 8.9 ± 10.8 vs. 12.9 ± 10.0; *p* = 0.022, respectively). The ODI and VAS scores at baseline, 1 month, and 1 year after surgery were not significantly different between groups S and L (Table 2). A comparison between the two groups of patients with only cauda equina symptoms (cauda equina syndrome plus mixed type) showed that the ODI and leg pain VAS scores at 3 months postoperatively were significantly lower in group S (19.9 ± 9.1 vs. 14.1 ± 12.5; *p* = 0.037, 7.4 ± 13.9 vs. 14.7 ± 23.1; *p* = 0.032, respectively). The ODI and VAS scores at baseline, 1 month, 6 months, and 1 year after surgery were not significantly different between groups S and L (Table 3).

There were no significant differences in patient satisfaction scores (satisfaction with surgery and present condition) 1 year after surgery between groups S and L (90.2 ± 14.1 vs. 89.3 ± 21.7; *p* = 0.760, 84.4 ± 21.2 vs. 84.5 ± 20.1; *p* = 0.983, respectively) (Table 4).

## 4. Discussion

This study investigated the effects of preoperative DOS on the clinical outcomes of MIS-TLIF. We found that patients with a DOS ≥1 year had a significant tendency for a high ODI score at 3 and 6 months after surgery compared with those with a DOS <1 year. In addition, patients with a DOS ≥1 year also had a significant tendency for high leg pain VAS scores 6 months after surgery. However, significantly, we observed that the ODI and VAS showed significant improvement regardless of the DOS 1 year after surgery. In addition, the patient satisfaction scores 1 year after surgery were not significantly different.

Lumbar degenerative disease with neurological symptoms is a major cause of disability and is commonly managed surgically when conservative management is exhausted. Surgical treatment for lumbar degenerative diseases is expected to provide short- or long-term pain relief and functional recovery [1]. However, the optimal timing for surgery remains uncertain. In the past, early surgical intervention was recommended for treating symptomatic LSS, based on the perspective that the disease is always progressive [12]. Kornblum et al. [13] recommended surgery for patients with 3 months of unsuccessful nonoperative treatment. Many previous studies have reported that patients with shorter DOSs before surgery have better clinical outcomes [3,11,14,15]. For example, Ng et al. [3] illustrated that patients with symptom durations of <33 months had more favorable outcomes for lumbar decompression surgery. Radcliff et al. [11] also concluded that patients with LSS < 12 months had better surgical outcomes. In addition, Jønsson et al. [14] reported that LSS patients with a preoperative duration of pain >4 years had inferior surgical outcomes at the 2-year follow-up. Nygaard et al. [15] also indicated that leg pain lasting >8 months correlated with unfavorable postoperative outcomes in patients with lumbar disc herniation. Conversely, similar to our findings, McGregor et al. [16] found no association between symptom duration and surgery outcomes at the 11-year follow-up. Additionally, Movassaghi et al. [5] reported that symptom chronicity did not significantly affect postoperative clinical outcomes or reoperation rates. However, these studies had several disadvantages. First, the surgical method was not fixed; therefore, there was a selection bias. Patients with degenerative spondylolisthesis, radiological evidence of instability, decompression of more than two levels, or patients aged <60 years were more likely to undergo decompression with instrumented fusion [3]. Second, the number of surgical levels was not constant. Finally, in most of these studies, conventional surgical methods were used for lumbar degenerative diseases, and no reports have been performed using MIS surgery alone. Therefore, we investigated the effect of the DOS on postoperative clinical outcomes and patient satisfaction in MIS-TLIF at a single level and found no significant difference in clinical outcomes 1 year after surgery.

One reason early surgical intervention is recommended is that prolonged compression of the cauda equina may cause nerve root ischemia and demyelination [17,18]. Simotas et al. [19] explained that the mechanism of neurogenic claudication is interpreted as cauda equina interruption of blood flow, venous congestion, ischemia, axonal damage, and intraneural fibrosis. From our results, this mechanism may have been attributed to the fact that improvement in the ODI was slower in group L. However, as shown in Figure 2, clinical outcomes, especially leg pain and numbness 1 year after surgery, were not significantly different. An experimental study using the porcine cauda equina showed that nerve root function recovery depends on the duration of compression [20]. Contrarily, this study suggests that a longer preoperative DOS may delay the recovery of physical function and neurological symptoms. However, the recovery is not poor 1 year after surgery, regardless of the ODI. Although it may delay the timing of surgical intervention, conservative treatment of lumbar degenerative disease is often successful. Furthermore, considering our results, we believe prolonged conservative treatment does not negatively affect surgical outcomes. Progressive paralysis is expected to have a poor clinical outcome, and we have a policy of early surgical treatment for such cases. However, in this study, there was no significant difference in the presence of preoperative paralysis between the two groups.

Regarding patient satisfaction, we evaluated two parameters of patient postoperative satisfaction: satisfaction with the surgery and satisfaction with the present condition. When asked about postoperative satisfaction, patients are often more focused on their current health status and symptoms than on how their symptoms improved after surgery. Hence, to evaluate patient satisfaction in more detail, patients were questioned separately about their satisfaction with the surgery and their present condition. We found that patients in the two groups met high levels of patient satisfaction (both with the surgery and present conditions) postoperatively. Additionally, there were no significant differences between the two groups. Few studies have evaluated the effect of the DOS on patient satisfaction. This study’s results agree with those of Movassaghi et al. [5], who treated 210 patients after lumbar decompression-only surgery and showed good satisfaction rates, regardless of the chronicity of symptoms. Similarly, Gaetani et al. [21] reported that patient satisfaction in 403 cases of herniated lumbar disc disease was not dependent on symptom duration. This is because satisfaction is most likely multifactorial and not necessarily associated with improvements in the outcome parameters [22]. For instance, preoperative expectations, psychological distress, preoperative health, and other intrinsic factors may contribute to patient satisfaction [23,24,25]. Therefore, in this study, the DOS did not appear to affect patient satisfaction significantly.

Our study had several limitations. First, the postoperative follow-up in our study may be insufficient to capture the long-term outcomes following spinal surgery. Patients with longer follow-up periods tend to have less favorable results, and studies with follow-up periods between 5 and 10 years have shown that 20–30% of the patients had unfavorable outcomes [26,27,28]. Our study was a pilot study; therefore, it is necessary to plan continuous follow-ups with these patients, prospectively, to evaluate the consistency of the results at a future long-term follow-up. Second, patients treated earlier were likely to have a higher disability, which may have added to the potential for selection bias. In addition, younger patients and patients with high activity may request early surgical intervention with the goal of early reintegration into society. However, the patient demographics at baseline were not significantly different between the two groups. Finally, we could not include mental status scores such as the SF-36. However, because the ODI has sections such as sleep and social life, it shows sufficient correlation with the SF-36’s physical and mental subscales [29]. Thus, the ODI may be a substitute for evaluating mental status.

## 5. Conclusions

We investigated the effect of preoperative DOS on the clinical outcomes of MIS-TLIF. The ODI scores tended to improve poorly in patients with a short DOS 3 or 6 months after surgery. However, clinical outcomes significantly improved 1 year after surgery, regardless of the DOS. Patient satisfaction was also excellent regardless of DOS. The results of this study suggest that prolonged conservative treatment does not negatively affect surgical outcomes.

## Figures and Tables

**Figure 1 medicina-59-00022-f001:**
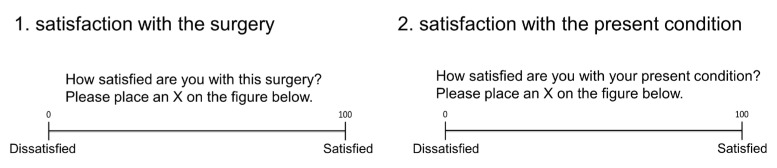
Methods of assessing patient satisfaction. Patient satisfaction was evaluated based on two parameters: satisfaction with surgery and satisfaction with the present condition, using the VAS (graded as ‘0’ for unsatisfied to ‘100’ points for very satisfied) at 1 year postoperatively. The survey was conducted by physiotherapists.

**Figure 2 medicina-59-00022-f002:**
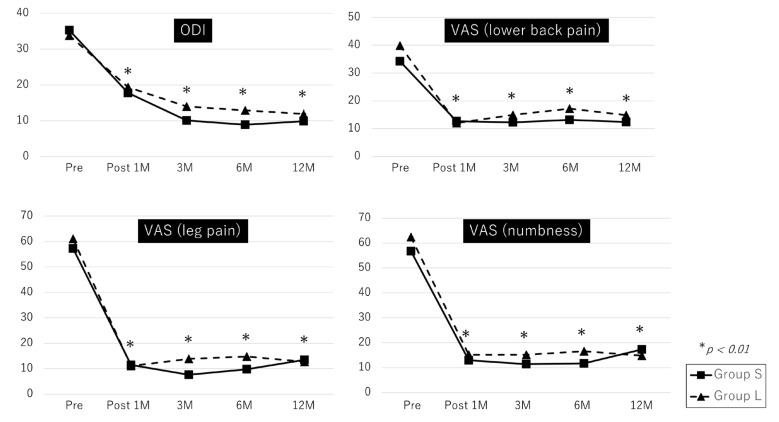
Comparison of preoperative and postoperative ODI scores, lower back pain, leg pain, and numbness VAS scores. All parameters showed significant improvement from the preoperative baseline to any time point after surgery. * *p* < 0.01; ODI, Oswestry Disability Index; VAS, Visual Analog Scale.

**Table 1 medicina-59-00022-t001:** Patient demographics.

		Total	Group S	Group L	*p* Value	95% CI
		163	60	103		
Sex (Female)		51.5%	45.0%	55.3%	0.21	−0.06, 0.26
Age		67.4 ± 8.9	66.6 ± 10.0	67.9 ± 8.2	0.40	−4.32, 1.74
BMI		24.0 ± 3.4	24.1 ± 3.3	23.9 ± 3.4	0.66	−0.84, 1.31
DM		16.0%	15.0%	16.5%	0.80	−0.13, 0.10
Smoking		22.3%	30.0%	19.4%	0.14	−0.36, 0.25
Preoperative paralysis		50.3%	43.3%	54.4%	0.18	−0.27, 0.50
Neurological symptom	Cauda equina syndrome plus Mixed Type	74.2%	68.3%	77.7%	0.21	−0.52, 0.24
	Radiculopathy	25.8%	31.7%	22.3%		
Disease	LSS (DS−)	35	19	16	0.14	
	LSS (DS+)	117	35	82		
	IS	11	6	5		
Surgical level	L2/3	3	1	2		
	L3/4	27	10	17		
	L4/5	115	39	76		
	L5/6	2	1	1		
	L5/S1	16	9	7		
Preoperative Radiographic parameters	PI-LL	11.1 ± 11.1	10.8 ± 11.0	11.3 ± 11.3	0.79	−4.15, 1.00
	PT	18.9 ± 8.3	17.9 ± 7.7	19.4 ± 8.5	0.23	−4.05, 3.09

Data are shown as the mean ± standard deviation; CI, Confidence Interval; DM, Diabetes Mellitus; LSS, Lumbar Spinal Stenosis; DS, Lumbar Degenerative Spondylolisthesis; IS, Isthmic Spondylolisthesis; PI, Pelvic Incidence; LL, Lumbar Lordosis; PT, Pelvic Tilt. In group S, the duration of symptoms (DOS) was <12 months; in group L, the DOS was ≥12 months.

**Table 2 medicina-59-00022-t002:** The comparison of clinical outcomes at preoperative and all follow-up periods (All patients).

	Group S	Group L	*p* Value	95% CI
Preoperative				
ODI	35.3 ± 16.7	33.8 ± 14.0	0.550	−3.54, 6.60
VAS (lower back pain)	34.2 ± 31.0	39.9 ± 28.1	0.242	−15.37, 3.92
VAS (leg pain)	57.3 ± 30.7	61.1 ± 27.1	0.440	−13.18, 5.77
VAS (numbness)	56.7 ± 31.6	62.4 ± 27.7	0.244	−15.46, 3.98
1 month after surgery				
ODI	17.8 ± 13.3	19.4 ± 13.5	0.470	−5.87, 2.72
VAS (lower back pain)	12.7 ± 18.7	12.1 ± 14.9	0.831	−4.99, 6.20
VAS (leg pain)	11.5 ± 20.2	11.1 ± 14.2	0.900	−5.50, 6.24
VAS (numbness)	13.0 ± 20.1	15.2 ± 20.6	0.503	−8.72, 4.30
3 months after surgery				
ODI	10.1 ± 8.9	14.0 ± 12.2	0.019 *	−7.21, −0.65
VAS (lower back pain)	12.3 ± 16.9	15.0 ± 21.4	0.386	−8.64, 3.36
VAS (leg pain)	7.6 ± 14.1	13.9 ± 21.9	0.027 *	−11.90, −0.74
VAS (numbness)	11.4 ± 19.9	15.2 ± 24.0	0.284	−10.66, 3.15
6 months after surgery				
ODI	8.9 ± 10.8	12.9 ± 10.0	0.022 *	−7.33, −0.58
VAS (lower back pain)	13.2 ± 20.0	17.2 ± 22.2	0.236	−10.70, 2.66
VAS (leg pain)	9.8 ± 19.8	14.8 ± 22.7	0.140	−11.77, 1.68
VAS (numbness)	11.7 ± 21.9	16.6 ± 26.5	0.210	−11.77, 1.68
1 year after surgery				
ODI	9.8 ± 9.2	11.9 ± 11.4	0.219	−5.26, 1.21
VAS (lower back pain)	12.4 ± 18.4	14.9 ± 20.1	0.405	−8.70, 3.54
VAS (leg pain)	13.5 ± 21.7	12.7 ± 20.7	0.798	−5.97, 7.74
VAS (numbness)	17.3 ± 27.7	14.8 ± 23.4	0.561	−5.95, 10.92

Data are shown as the mean ± standard deviation; CI, Confidence Interval; ODI, Oswestry Disability Index; VAS, Visual Analog Scale *, *p* < 0.05; In group S, the duration of symptoms (DOS) was <12 months; in group L, the DOS was ≥12 months.

**Table 3 medicina-59-00022-t003:** The comparison of clinical outcomes at preoperative and all follow-up periods (only the patients with cauda equina syndrome plus mixed type).

	Group S	Group L	*p* Value	95% CI
Preoperative				
ODI	347 ± 17.1	33.6 ± 14.6	0.73	−5.14, 7.31
VAS (lower back pain)	33.7 ± 29.9	36.2 ± 28.1	0.66	−13.69, 8.70
VAS (leg pain)	52.8 ± 30.8	59.9 ± 27.9	0.22	−18.47, 4.36
VAS (numbness)	59.2 ± 30.3	62.4 ± 27.4	0.58	−14.41, 8.08
1 month after surgery				
ODI	17.1 ± 13.3	19.1 ± 13.3	0.440	−7.07, 3.10
VAS (lower back pain)	14.5 ± 20.4	12.4 ± 15.3	0.589	−5.47, 9.55
VAS (leg pain)	12.9 ± 23.3	11.9 ± 15.1	0.801	−7.00, 9.03
VAS (numbness)	15.2 ± 22.8	15.0 ± 20.8	0.959	−8.25, 8.69
3 months after surgery				
ODI	9.9 ± 9.1	14.1 ± 12.5	0.037 *	−8.17, −0.27
VAS (lower back pain)	13.8 ± 17.3	14.8 ± 21.6	0.782	−8.20, 6.18
VAS (leg pain)	7.4 ± 13.9	14.7 ± 23.1	0.032 *	−13.97, −0.62
VAS (numbness)	11.5 ± 20.5	16.8 ± 26.0	0.216	−13.94, 3.19
6 months after surgery				
ODI	9.8 ± 12.1	12.9 ± 10.4	0.166	−7.53, −1.32
VAS (lower back pain)	14.3 ± 20.6	16.6 ± 21.4	0.571	−10.25, 5.69
VAS (leg pain)	10.8 ± 20.8	14.8 ± 23.8	0.345	−12.35, 4.36
VAS (numbness)	12.2 ± 22.4	18.1 ± 28.4	0.210	−15.33, 3.42
1 year after surgery				
ODI	9.1 ± 9.5	11.7 ± 11.8	0.193	−6.53, 1.34
VAS (lower back pain)	13.0 ± 18.3	14.9 ± 20.5	0.588	−9.25, 5.28
VAS (leg pain)	13.3 ± 23.4	13.2 ± 21.4	0.991	−8.65, 8.75
VAS (numbness)	19.5 ± 30.3	15.6 ± 24.8	0.470	−6.96, 14.92

Data are shown as the mean ± standard deviation; CI, Confidence Interval; ODI, Oswestry Disability Index; VAS, Visual Analog Scale *, *p* < 0.05; In group S, the duration of symptoms (DOS) was <12 months; in group L, the DOS was ≥12 months.

**Table 4 medicina-59-00022-t004:** The comparison of patient satisfaction scores.

	Group S	Group L	*p* Value	95% CI
Satisfaction with the surgery	90.2 ± 14.1	89.3 ± 21.7	0.760	−4.70, 6.41
Satisfaction with the present condition	84.4 ± 21.2	84.5 ± 20.1	0.983	−6.76, 6.61

Data are shown as the mean ± standard deviation; CI, Confidence Interval; group S, the duration of symptoms (DOS) was <12 months; group L, the DOS was ≥12 months.

## Data Availability

Data collected for this study, including individual patient data, will not be made available.

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
