# Peer review of "Influence of the Preoperative Duration of Symptoms on Patients’ Clinical Outcomes after Minimally Invasive Surgery-Transforaminal Lumbar Interbody Fusion for Degenerative Lumbar Spinal Diseases"

_medicina, 2022, doi:10.3390/medicina59010022_

Round 1

Reviewer 1 Report

We read with great interest this paper, which aimed to ascertain the effect of preoperative DOS on the surgical outcomes of MIS-TLIF over a 1-year follow-up period and concluded that prolonged conservative treatment does not negatively affect surgical outcomes.

However, there are several concerns with the acceptance of this paper.

Major

1)    I think you need a preoperative in the title exactly.

2)    Although types of neurological symptoms (cauda equina syndrome, mixed type, or radiculopathy) of the two groups were evaluated , there was only radiculopathy listed.

 Was there a significant difference?

Since, as the author pointed out, cauda equina disorders tend to remain symptomatic.

  Authors should list preoperative and postoperative by neurological symptoms

3)> The ODI and VAS changes were defined as preoperative ODI minus postoperative ODI and preoperative VAS minus postoperative VAS, respectively.

Is this a methodologically correct analysis? Please provide references.

4)>Preoperatively, there were no significant differences in the baseline patient demographics between groups S and L  (Table 1).

   Is there any difference in the breakdown of Disease(DS+-,IS) between the two groups? If there is no significant difference, it should be stated.

Minor

1)    Is preoperative conservative treatment a standardized protocol (no difference between the two groups?) Usually, the conservative treatment depends on the neurological symptoms. The conservative treatment may have affected the outcome.

2)    Did the degree of slip correction differ between the two groups?

Round 2

Reviewer 2 Report

I think these changes are sufficient. Thank you!